# Microbial Contamination and Survival Rate on Different Types of Banknotes

**DOI:** 10.3390/ijerph19074310

**Published:** 2022-04-04

**Authors:** Derniza Cozorici, Roxana-Alexandra Măciucă, Costel Stancu, Bianca-Maria Tihăuan, Robert Bogdan Uță, Cosmin Iulian Codrea, Răzvan Matache, Cristian-Emilian Pop, Robert Wolff, Sergiu Fendrihan

**Affiliations:** 1Non-Governmental Research Organization Biologic, 14 Schitului Str., 032044 Bucharest, Romania or costelstancu@ngobiologic.com (C.S.); robertuta@ngobiologic.com (R.B.U.); fendrihan.sergiu@ngobiologic.com (S.F.); 2Department of Bioengineering and Biotechnologies, Faculty of Medical Engineering, University Politehnica of Bucharest, 011061 Bucharest, Romania; derniza.cozorici@stud.fim.upb.ro; 3Faculty of Biology, University of Bucharest, 91–95 Splaiul Independenței, 050095 Bucharest, Romania; maciuca.roxana-alexandra@s.bio.unibuc.ro; 4Research Institute of the University of Bucharest—ICUB, 91-95 Splaiul Independenței, 050567 Bucharest, Romania; bianca.tihauan@sanimed.ro; 5Research & Development for Advanced Biotechnologies and Medical Devices, SC Sanimed International Impex SRL, 087040 Călugăreni, Romania; 6“Ilie Murgulescu” Institute of Physical Chemistry, Romanian Academy, 202 Splaiul Independentei, 060021 Bucharest, Romania; cosmin.codrea@stud.chimie.upb.ro; 7Department of Science and Engineering of Oxide Materials and Nanomaterials, Faculty of Applied Chemistry and Materials Science, University Politehnica of Bucharest, 060042 Bucharest, Romania; 8National Institute for Research and Development in Environmental Protection, 294 Splaiul Independentei, 060031 Bucharest, Romania; matache.razvan@incdpm.ro; 9Department of Biochemistry and Molecular Biology, Faculty of Biology, University of Bucharest, 91-95 Splaiul Independenței, 050095 Bucharest, Romania; 10College of Nursing and Public Health, South University, 9 Science Ct., Columbia, SC 29203, USA; rwolff@southuniversity.edu; 11Faculty of Medicine, “Vasile Goldis” University, Revoluției Blvd. 94, 310025 Arad, Romania

**Keywords:** public health, banknotes, currency, pathogens, fomite, bacterial adherence

## Abstract

In the COVID-19 pandemic context, numerous concerns have been raised regarding the hygienic status of certain objects we interact with on a daily basis, and especially cash money and their potential to harbor and transmit pathogenic bacteria. Therefore, in the present study, we analyzed different currency bills represented by British pounds (5 £, 10 £ and 20 £), Romanian lei (1 leu, 5 lei and 10 lei), U.S. dollars (1 $, 5 $ and 10 $) and Euros (5 €, 10 € and 20 €) in order to evaluate the bacterial survival rate and bacterial adherence. We used five reference microorganisms by American Type Culture Collection (ATCC, Manassas, VA, USA): *Staphylococcus aureus* ATCC 6538, *Escherichia coli* ATCC 8739, *Enterococcus* sp. ATCC 19952, *Salmonella enterica* subsp. enterica serovar *Typhi* ATCC 6539, and *Listeria monocytogenes* ATCC 7644. Microorganisms were selected in accordance with the criteria of prevalence, pathogenicity, opportunism, and incidence. However, Maldi-TOF analysis from samples taken from the banknotes revealed only a few of the common pathogens that are traditionally thought to be found on banknotes. Some of the most important factors for the survival of pathogenic agents on surfaces are the presence of organic matter, temperature and humidity. Our data showed that *Salmonella enterica* survived 72 h on every banknote tested, while *L. monocytogenes* tended to improve persistence in humid conditions. Survival rate is also influenced by the substrate composition, being lower for polymer-based banknotes especially for *Salmonella enterica*, *Listeria monocytogenes* and *Enterococcus* sp. The adherence of bacterial strains was lower for polymer-based banknotes British pounds and Romanian Leu, in contrast to the cotton-based U.S dollars and Euro banknotes. The risk of bacterial contamination from the banknote bills is high as indicated by both a strong survival capacity and low adherence of tested bacteria with differences between the two types of materials used for the tested banknotes.

## 1. Introduction

Banknotes, often called paper currency, are generally considered to be one of the most common fomites that people handle. Because they are frequently in contact with, or exposed to, numerous individuals, surfaces (objects), food and aerosols, they may be responsible for the spread of many diseases.

Banknotes mainly fall into two large groups, paper banknotes, actually manufactured from cotton fiber, and the plastic (or polymer) banknotes. The cotton used for the paper banknotes is sometimes mixed with linen, abaca, or other textile fibers and infused with polyvinyl alcohol or gelatin. In line with this, our research is focused on determining if the material of which the banknotes are manufactured is a factor that influences the bacterial survival rate and the bacterial adherence to the banknotes.

The euro banknotes are printed on 100% cotton paper, which lends them a unique crispness and wear resistance [1]. Cotton fibers are the purest form of cellulose, the most prevalent polymer in nature, which is found in plant cell walls. The cotton fibers are made up of nearly 90% cellulose and other non-cellulosic compounds such as proteins, waxes, pectins and other substances. These non-cellulose elements are eliminated during chemical processing, and the cellulose content of cotton fibers increases to more than 99% [2]. Cotton cellulose has a higher degree of polymerization and crystallization than wood cellulose used in regular paper, leading to stronger fibers.

US dollar banknotes are manufactured of a specifically developed cotton-linen blend provided solely by the Crane Company. The composition has evolved over time, and today’s US paper currency is composed of 75% cotton and 25% linen, according to the Bureau of Engraving and Printing [3]. Linen is a natural, multicellular, and cellulosic fiber, which is obtained from flax plant stem fibers. It is well known that linen is one of the strongest fibers available, with a tensile strength greater than cotton. It gives paper more strength and durability. Color, strength and sizing agents are also added to the raw materials during the papermaking process [4,5].

Romania and the United Kingdom, on the other hand, employ synthetic polymer substrates for their banknotes. The polymer used to manufacture the Romanian leu and the British pound sterling is biaxially oriented polypropylene (BOPP). Biaxially oriented polypropylene or BOPP film is a thin plastic film that has been stretched in two directions, longitudinally and transversely. The resultant material has a high tensile strength, clarity, and chemical resistance. BOPP is a non-fibrous and non-porous material, making banknotes produced of it harder to rip, more resistant to folding, more resistant to soil, waterproof, harder to burn and easier to machine process than paper banknotes. Banknotes made using BOPP are also recyclable at the end of their lives [6,7]. Because the polymer is impermeable and non-fibrous, meaning it repels dirt and moisture, the polymer banknotes are expected to be cleaner than those made of cotton paper [8]. Prior to printing, the polymeric substrate is covered with several layers of thick white ink to provide a surface to which the printing ink is more likely to adhere. A protective coat of polyurethane varnish is added to the banknote’s surface after printing and the implementation of security elements. Ground silica (SiO_2_) is also used to keep the surface from getting overly slippery, which would cause the banknotes to stick together [4].

Ink materials are one of the most significant components of a banknote, especially in terms of anti-counterfeiting and anti-alteration protection. Because they are not widely available and the exact formulation of the inks used in banknote printing is a highly guarded secret, all of the inks used in banknote printing provide a certain level of security. The inks used must be high-performing and resistant to external factors including light, heat, and moisture [9]. Fluorescent ink, metameric ink, magnetic ink, copy-proof ink and optically variable ink are only a few examples of anti-counterfeiting inks.

The ink used to print U.S. paper banknotes is made up of specific formulations developed by the U.S. Treasury. Green ink is printed on the reverse of each bill and depending on the bill, several inks are used on the front, such as black ink, color-shifting ink and metallic ink [3]. Color-shifting ink or optical variable ink is an ink that contains optically variable pigments that change color based on light incidence and angle of view. A layered optical interference structure is one of the most frequent types of optically variable pigment. At least one metallic reflecting layer, one transparent dielectric layer and one semitransparent metal layer are all common components of an interference structure. The reflecting layer is made of metals like aluminum, gold, copper, or silver, while the transparent dielectric layer is made of chemical compounds like magnesium fluoride, silicon dioxide, or aluminum oxide, and the semitransparent metal layer is made of metals like chromium or nickel [10,11].

Magnetic ink, which contains colloidal particles made up of iron (II) and iron (III) compounds like FeCl_2_ and FeCl_3_, is also used on U.S. banknotes. The components can be mixed into any color of ink or colorless varnishes. Unique magnetic sensors or special gadgets can identify or visualize the text or image created by the ink [12].

The color scheme for euro banknotes includes grey for the €5 note, and red and blue for the €10 and €20 notes [13]. UV fluorescent ink is one type of ink utilized in the production of the most frequently used euro banknotes. UV fluorescent ink is a type of ink that contains fluorescent pigments that glows when exposed to UV light. For the most frequently used Romanian banknotes: 1, 5 and 10 lei, iridescent ink, UV fluorescent ink for the numerical value and series of banknotes and magnetic ink are used [14]. A metallic pigment formed up as ink and printed on top of the polymer substrate as a silver patch is applied on the £5, £10 and £20 notes, in addition to UV fluorescent ink [10,15].

Contaminated banknotes can play a role in the transmission of microorganisms between people, as they are passed from hand to hand, especially in the countries where digital options are not yet widespread. Banknotes are indicated as a vehicle of possible contamination with pathogenic bacteria; therefore, in the present study, we analyzed different currency bills from the two large groups, paper banknotes, manufactured from cotton fiber, and the plastic polymer banknotes. We suspect the material of which the banknotes are manufactured to be a factor that influence the bacterial survival rate and the bacterial adherence on the banknotes.

The current COVID-19 pandemic has brought hygiene to the forefront of concerns, including the spread of the SARS-CoV-2 virus, flu viruses, and the increase in antibiotic resistant bacteria that might be spread via fomites and transferred from banknotes to skin [16,17,18]. These bacteria from banknotes can be transferred to human skin [18]. For better understanding, it is important to know that bacteria and viruses often occupy the same niches in the human body. There is no direct physical interaction between the virus and the bacteria; rather, the viral infection makes one or more host cell types more susceptible to bacterial colonization [19]. That is why it is important to include both in a discussion about potential infections that can be acquired by money handling. Bacteria and viruses are able to survive for long periods on fomites or inanimate objects, including banknotes or paper currency [18,20]. Gedik et al. [18] showed some increased survival rates of bacteria on plastic-based banknotes over cotton based, but the results were inconsistent. Wißmann et al. [20] found many studies and overall showed greater survival of bacteria on plastics over various cloth materials, though there was not a focus on banknotes.

In contrast, it was found that bacteria on a plastic polymer surface has lower adherence on this material and a lower survival rate than bacteria on banknotes with cotton or cotton/linen blends [21].

This study examines both the survival and adherence of common pathogenic bacteria on various banknote types and denominations used commonly in the US, UK, Romania, and the Eurozone of the EU. This contribution is to add to the knowledge of these banknotes serving as fomites in the transmission of bacteria and to highlight the difference between the banknote’s material regarding this characteristic.

## 2. Materials and Methods

### 2.1. Sampling Rationale for Preliminary Microflora Identification

Testing samples represented by 12 banknotes (British pounds—5 £, 10 £ and 20 £, Romanian lei—1 leu, 5 lei and 10 lei, U.S. dollars—1 $, 5 $ and 10 $ and Euros—5 €, 10 € and 20 €) were obtained from an exchange office, manipulated with sterile gloves, and stored before testing in sealed sterile pouches. We investigated different denomination bills from each type of currency keeping in mind different circulation speeds given their value, convenience of use, and the number of banknotes issued by each national bank.

Samples used for matrix-assisted laser desorption/ionization-time of flight (MALDI-TOF) analysis were collected from each banknote from a surface area of 5 cm^2^ on each side, using dry swabs (Copan eSwab, Brescia, Italy). For primary identification of microorganisms, we used Columbia blood agar, Chocolate agar (CHOC), Cystine–lactose–electrolyte-deficient agar (Cled), and Sabouraud agar (Oxoid, Thermofisher, France). Final taxonomic identification was performed with a MALDI-TOF Biotyper (MALDI-TOF Microflex LT/SH, BrukerDaltonik, Bremen, Germany). From isolated colonies, MSP 96 target polished steel plate (Bruker Daltonics, Germany, Bremen) spots were loaded, and on each spot α- Cyano-4-hydroxycinnamic acid (Bruker Daltonics, Bremen, Germany) matrix was added among the stock solution (acetonitrile, trifluoracetic acid and distilled water). The SR Library Database was used for the identification (Database CD BT BTyp2.0-Sec-Library 1.0) with MBT Compass (v.4.1) software.

### 2.2. Assessment of Bacterial Survival Rate

Testing samples described in Section 2.1 were first decontaminated by immersion in ethanol solution 70% for 15 min and then followed by 6 h of exposure to UV light in a class III biosafety cabinet, in order to ensure a sterile surface of bills. Assessment of survival rate was performed using five reference microorganisms by American Type Culture Collection (ATCC, Manassas, VA, USA): *Staphylococcus aureus* ATCC 6538, *Escherichia coli* ATCC 8739, *Enterococcus* sp. ATCC 19952, *Salmonella enterica* subsp. enterica serovar *Typhi* ATCC 6539, *Listeria monocytogenes* ATCC 7644. The main criteria of microorganisms’ selection were prevalence, pathogenicity, opportunism, and incidence [22,23,24]. In addition, 1.5 × 10^8^ CFU/mL microbial suspensions (adjusted to standard density of 0.5 McFarland) were obtained from fresh 15 to 18 h cultures, developed on solid medium Mueller–Hinton agar (Biomerieux, France) and then serially diluted to 10^5^ CFU/mL using a saline solution. On every currency bill, squares of 1 × 1 cm^2^ were delimited. Furthermore, 10 µL of bacterial suspension were spread out on each square using a Drigalski spatula. The inoculated bills were incubated in two conditions: (a) in an incubator at 36 ± 2 °C in sterile Petri dishes and, (b) in a closed biosafety cabinet at 22 ± 2 °C, 35% humidity, then evaluated at 0, 12, 24, 48 and 72 h by plating on solid medium Mueller–Hinton agar the designated square for each time point using a swab moistened in phosphate saline solution. A negative control represented by saline buffer was used for every time point.

In order to avoid cross contamination, all currency bills were evaluated for one microorganism at time and then ran through the same decontamination procedure. After incubation, plates were read by counting the colonies. All tests were performed in triplicate. Rate of survival was expressed as logarithmic reduction. The logarithmic reduction was calculated using the formula: Reduction (CFU/mL) = lgA − lgB, where A is CFU/mL of positive control (initial number of colony forming units in the inoculum); and B is CFU/mL of samples (number of colony forming units at x time point).

### 2.3. Evaluation of Bacterial Adherence

The inert surfaces of the banknotes were tested for their capacity to inhibit bacterial adhesion using the five reference microorganisms mentioned in Section 2.2. The samples (decontaminated using the method described in Section 2.2) were immersed in 9 mL of Brain Heart Infusion (BHI) medium (Biomerieux, Marcy-l’Étoile, France) seeded with 1 mL of 10^5^ CFU/mL obtained by diluting in a saline buffer the suspension 0.5 McFarland—1.5 × 10^8^ CFU/mL, and incubated at 36 ± 2 °C for 4 h. The suspensions used were checked by a plate counting method.

After the contact with the microbial inoculum, the banknotes were dried out in a biosafety cabinet. After 24 h from the initial inoculation, the dried banknotes were washed in a phosphate buffer solution (PBS) in order to remove non-adherent microorganisms and then immersed in a saline buffer and shaken vigorously using a vortex (Thermo Scientific, Waltham, MA, USA) in order to extract adherent microorganisms. The microbial adherence was assessed spectrophotometrically, by measuring the absorbance at λ = 620 using FlexStation 3 (Molecular Devices Company, Sunnyvale, CA, USA). In order to assess the percent of adherent versus non-adherence bacterial cells, we compared the optical density of the initial 10^5^ CFU/mL inoculum to the saline buffer solution obtained after vortexing the banknotes and subtracting the absorbance of the PBS solution used to initially wash bills.

### 2.4. Statistical Analysis

The statistical analysis was performed using GraphPad Prism 9 (San Diego, CA, USA). Data were analyzed using the one-way ANOVA test. The level of significance was set to *p* < 0.05.

## 3. Results and Discussion

As the COVID-19 pandemic came unexpectedly for the general public, numerous concerns have been raised regarding the hygienic status of certain objects we interact with on a daily basis, and especially cash money and its potential to harbor and transmit pathogenic bacteria. We believe that this particular interest peaked its values in the last two years, but, as we discovered, people’s interest for the presence of microorganisms on currency started in the late 1800s. Vriesekoop et al. mentioned roughly 100 publications that report on the presence of microorganisms on banknotes and/or coins since the beginning of the 19th century, with more than two-thirds of those papers being published since 2000 [21].

This considerable interest in the hygienic status of currency as well as the reported presence of pathogenic and opportunistic bacteria on banknotes, which tend to act as fomites, led us to focus this section of our study on analyzing the survival rate of specific microorganisms in various conditions, as close to real-life situations as possible.

Previously reported results show that the bacterial load on banknotes is dependent on the polymer content; therefore, a lower bacterial load was observed on polymer banknotes compared to cotton-based banknotes [25,26]. Samples analyzed in the present study are represented by British pounds (5 £, 10 £ and 20 £), Romanian lei (1 leu, 5 lei and 10 lei), U.S. dollars (1 $, 5 $ and 10 $), and euro (5 €, 10 € and 20 €). Dollar and euro bills are cotton-based banknotes, while pounds and Romanian lei are polymer-based.

The data obtained from the MALDI-TOF analysis regarding the identification of banknotes preliminary microflora (Table 1) revealed an array of Gram-positive microorganisms. The results were marked by the MALDI-TOF Software as high-confidence identification, meaning that an identification in which the species is identical to the best match was obtained. Among identified species, we distinguish some endospore-forming bacteria such as *Bacillus megaterium* and *Bacillus idriensis,* the last one on them being a relatively recently distinguished microorganism [27]. Although their variety is not one of impressive matters, with a low number of cells for some samples, their etiology is far more interesting. The microorganisms identified represent species of microorganisms associated with normal skin microbiota (e.g., *Staphylococcus epidermidis* is a major inhabitant of the skin, *Micrococcus luteus* not as commune as staphylococci but still present in normal skin microbiota) [28,29]. The *Bacillus* species are found in diverse habitats (such as soil, water, milk, honey, dried food) [27,30,31,32], and as they tend to colonize the body and by nature exhibit extended survival for long periods of time under scars conditions, their presence on banknotes is not surprising. Moreover, because they are spore-forming bacteria, the transmission rate is heightened. As for *Exiguobacterium aurantiacum* and *Fictibacillus arsenicus*, they can be found in different environmental niches such as industrial waste, metal ores, hot springs, freshwater, marine sediments, and soil, with a high human interaction rate; therefore, the chance of colonization arises [31,32,33].

The presence of this type of bacteria becomes a problem especially when an immunocompromised patient encounters soiled banknotes and especially when the act is not followed through with hygienic precautions. This type of commensal bacteria can be opportunistic, as when the medium is right, they can turn against their host provoking infections.

Although the identified microorganisms present potential health risks, we upped the ante by assessing the survival rate and adherence capacity of five organisms (*Staphylococcus aureus*, *Escherichia coli, Enterococcus* sp., *Salmonella enterica* subsp. *enterica* serovar *Typhi* and *Listeria monocytogenes*) with higher rates of infection if contacted [34] and reported their presence on similar types of banknotes [26,35,36].

There are a multitude of key factors that contribute to the successful survival of microorganisms, the main one being the varying degrees of stability within the environment. For the most part, bacterial population dynamics are controlled by nutrient sources and environmental conditions (oxygen concentration, nutrient levels, temperature, pH, etc.) [37]. Therefore, for the assessment of survival rate, bacteria extraction from the inoculated surfaces was performed as close as possible to how the normal handling of banknotes goes. A sterile swab was immersed in saline solution and then drained by rolling it on a sterile Petri dish, in an effort to resemble the natural moisture of fingers. As we know, once the initial inoculum is dried/absorbed into the surface of banknotes, the microorganisms tend to migrate into the fibers; therefore, this assessment has its limitations.

The length of bacterial survival in the environment determines its relative prevalence and associated virulence traits. The survival rate at 22 °C (Figure 1) was observed to be strain specific, with a decreasing viability rate as time elapsed, and possible resources being consumed. In tested conditions (with 35% humidity), *Staphylococcus aureus* strains maintain high rates of viability, results that correlate with other literature reports [38,39]. R. H. Katzenberger et al. (2021) observed the impact of high humidity in correlation with increased viability, especially on *S. aureus* strains.

Our findings confirmed that there was an increased survival rate with humidity as we observed this at 37 °C and 65% humidity (Figure 2). Previous studies reported that the survival rate for bacteria such as *Escherichia coli* is dependent on surface nature and concentration of the initial inoculum, and can be extended from a few hours to over 30 days [40]. It is not considered an airborne pathogen, so, in this case, unwashed, or improperly washed, hands represent its main means of entry or contamination. The same goes for *Enterococcus* sp., with survival rates even higher, from 5 days up to 60 days, especially for vancomycin-resistant *Enterococcus* sp. [41].

All five bacterial strains demonstrated high survival rates for 72 h based on colony growth. For *Salmonella enterica* the results correlate with previous findings in the literature that show that *Salmonella* is more efficient at surviving in secondary habitats (i.e., outside the animal host) as compared to *E. coli.* Survival of *E. coli* in secondary habitats requires the ability to overcome low nutrient availability and temperature fluctuations [42].

*Listeria monocytogenes* is ubiquitous in nature due to its inherent ability to survive and grow under a wide range of adverse environmental conditions, such as low temperatures, high acidity and salinity, and reduced water activity. As most bacterial strains do, *L. monocytogenes* tends to be able to increase its persistence in humid conditions (Figure 3) [43], being able to survive from one day to months. Staphylococcus aureus had the highest survival rate of the bacterial species tested at 22 °C which is expected due to its common occurrence on the surface of the skin which has a lower temperature than the internal environment. It had the highest survival rates (Figure 1) due to its ability to survival in dry or higher saline conditions.

We set the time limit for testing for 72 h because, in the case of small banknotes (as the ones we tested), the passage to more handlers is more quickly compared to larger banknotes, which tend to be less utilized.

Some of the most important factors affecting the survival of pathogenic organisms on surfaces are the presence of organic matter, solar irradiation, temperature, and humidity. Previous studies show that the denomination and age of a bill directly correlates with contamination levels [44,45], and lower denominations were considered to be the most contaminated, presumably because lower denomination bills pass through more hands in their lifetime than the higher denomination bills.

By comparison, the survival rate of selected microorganisms (at 37 °C) is dependent on substrate composition (Figure 4), results consistent with previous research [25,26]. Therefore, for polymer-based banknotes, we observed a lower rate of bacterial survival, especially for *Salmonella enterica*, *Listeria monocytogenes* and *Enterococcus* sp. On the other hand, for cotton-based banknotes, higher survival rates were determined. Similar results were found [25] while studying Australian and New Zeeland currency that are polymer-based compared with Mexico currency, which is cotton-linen-based.

The comparison of logarithmic reduction of survival rate at 12 h vs. 72 h (Figure 5) follows a pattern, which is strain specific. Log reductions of survival rates were modest, no greater than 0.2 lg for *Staphylococcus aureus*, 0.13 for *Escherichia coli*, 0.12 for *Enterococcus* sp., 0.19 for *Salmonella enterica* and 0.05 for *Listeria monocytogenes.* As previous scientific reports mention, a longer period of time under harsh conditions is required for an ample reduction of viability.

The bacterial adherence pattern on the currency (Figure 6) indicates a high recovery rate of bacteria after the banknotes were washed in PBS after drying and then vigorously shaken in a saline buffer, using a vortex. The results indicate that this method managed to extract most of the initial bacterial inoculum. All banknotes were inoculated, one at a time, with *Staphylococcus aureus*, *Escherichia coli*, *Enterococcus* sp., *Salmonella enterica*, or *Listeria monocytogenes* suspensions of 10^5^ CFU/mL and allowed to sit for four hours. After drying them out for 24 h in a biosafety cabinet, and completing the shaking extraction, the optical density of the initial inoculum was compared with the optical density of the extraction media. These results indicate the low capacity of tested bills to maintain the adherence of the bacteria, especially for polymer-based ones. This occurred for strains under such conditions, thus offering possible solutions for efficient reduction in contamination of banknotes. Recovery percentages were over 70% with all strains recovered from every banknote. The highest recovery percentage was observed on banknotes inoculated with *Salmonella enterica* (over 85%) and the lowest on the ones inoculated with *E. coli* (70%).

Results obtained for the assessment of bacterial adherence capacity on banknotes correlates with other research findings [21,25,46] relating to banknote material content.

As mentioned above, the nature of banknotes manufacturing material plays an important role in the bacterial adherence process. In Figure 7, a clear pattern can be distinguished; hence, adherence of bacterial strains, either Gram negative (*Escherichia coli, Salmonella enterica*) or Gram positive (*Staphylococcus aureus, Enterococcus* sp., *Listeria monocytogenes*), was reduced (scoring high values of absorbance) on British pounds and Romanian leu, which as we know are polymer-based banknotes. This result correlates with the high recovery rates obtained. Even though the Romanian Leu should in theory exhibit low bacterial adherence similar to the British pound due to its polymeric base [18], a higher adherence rate (lower values of absorbance) was observed for *S. aureus* and *Salmonella enterica* strains. As for cotton and linen-based banknotes (euro and dollar), high adherence was observed on all tested strains, and there were significant differences in adherence between strains.

There is a clear difference in adherence patterns between banknotes dictated by their composition, their manufacturing materials and also the limited possibilities of reducing contamination. This should be especially true for cotton-based banknotes. Even if the scientific literature suggests an array of methods for cleaning, and disinfection such as UV exposure (sun exposure), periodical withdrawal of old and unsanitary currency [44], irradiation [45] and introduction for use of less adherent banknotes may be needed to reduce the transmission of disease organisms.

## 4. Conclusions

For the most part, there is little information regarding if and how often banknotes are cleaned. While it is assumed that some banks conduct some cleaning, we are currently unaware of how banknotes are cleaned. Some money sorting or packing equipment may use heat to help decontaminate the currency [47]. It is presumed that banknotes are generally not cleaned as they pass through numerous hands, businesses, banks and even traveling far distances. We do know that, globally, governments spend large amounts of resources annually for replacing soiled or damaged cash. In 2019, the NCBs (National Central Banks) categorized some 5.1 billion banknotes as unfit for circulation and replaced them [48].

A low number of bacterial strains with scarce diversity were collected from banknotes owing to the sampling methodology shortcomings or bills procurement source. The higher numbers registered for the local currency may simply indicate greater use and more recent handling. Survival of the various bacterial species tested was high on the banknotes, indicating that under the experimental conditions the viability of the bacteria was maintained more than some other studies. The decline in survival rates over time, up to 72 h, was expected but not as steep as previous studies. The higher survival rates at 37 °C compared to 22 °C may be due to all of the tested organisms being pathogens. A comparison to environmental bacteria might show a different pattern.

The adherence to the currency did not appear to be strong in the test conditions. Most of the bacteria were recovered (recovery rates of minimum 70%) from the notes following vigorous vortexing. There is a difference in adherence patterns between banknotes based upon their composition, with lower adherence manifested on the plastic polymer banknotes.

Banknotes manufactured with cotton or linen fibers (such as euro and dollar bills) present increased spaces in the surface texture and moisture retention capacity compared to polymer-based ones to increase retention or adherence. These may account for preventing the release of bacteria from its adherence to the banknote surface under study conditions. A critical question that still needs to be answered is whether a strong adherence keeps the bacteria available on the currency for higher levels of transfer to human skin, or does a lower adherence increase the risk of transfer to skin and objects? If adherence is lower, does this increase loss of the bacteria into the air during handling or does it increase transfer to skin and objects from the currency surface?

It appears from our data that the risk of bacterial contamination from the banknote bills indicates both a strong survival capacity and that the bacteria may easily be dislodged and transferred to skin or other objects. The role of ‘paper’ money, or banknotes, is still poorly understood in its potential role of transmission of pathogens. To suppress bacterial contamination and potential spread of disease, it will come down to hygienic self-care and self-awareness regarding money handling.

## Figures and Tables

**Figure 1 ijerph-19-04310-f001:**
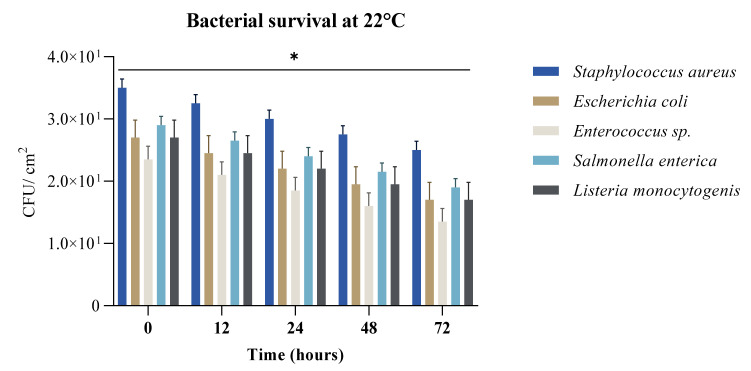
Bacterial survival recorded as CFU/cm^2^ at 22 °C, 35% humidity, on various banknotes at five time points (0 h, 12 h, 24 h, 48 h and 72 h) (* *p*-value = 0.03).

**Figure 2 ijerph-19-04310-f002:**
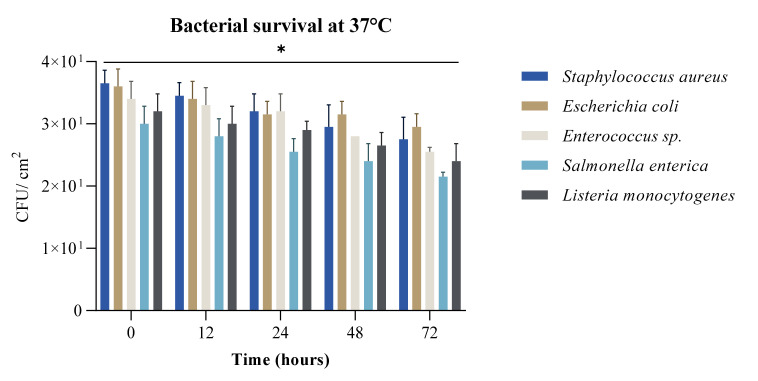
Bacterial survival recorded as CFU/cm2 at 37 °C, 65% humidity, on various banknotes at five time points (0 h, 12 h, 24 h, 48 h and 72 h) (* *p*-value = 0.02).

**Figure 3 ijerph-19-04310-f003:**
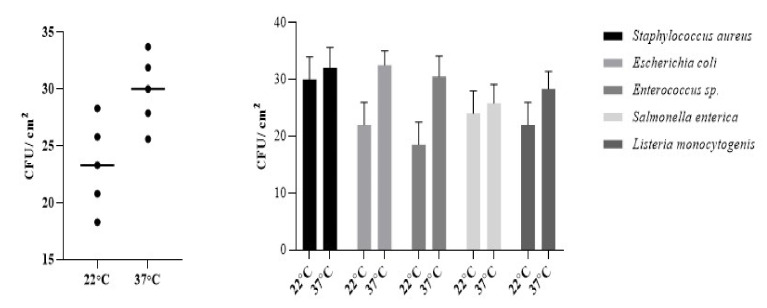
Influence of temperature of incubation on survival of different bacterial strains.

**Figure 4 ijerph-19-04310-f004:**
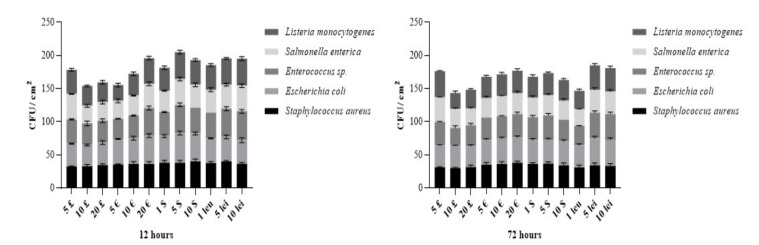
Bacterial survival rate on different banknotes—Comparison of 12 h after inoculation to 72 h post inoculation.

**Figure 5 ijerph-19-04310-f005:**
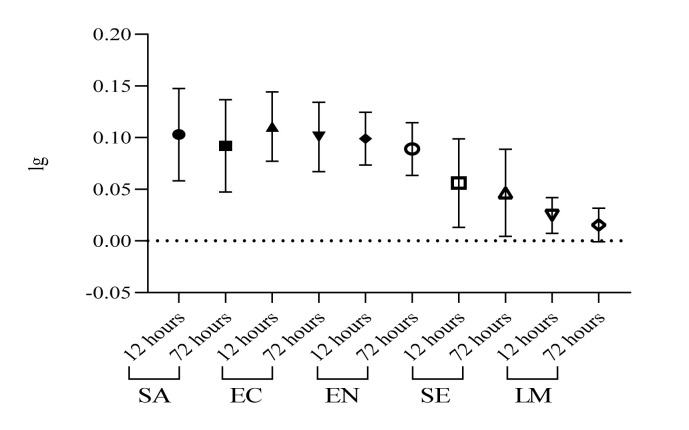
Logarithmic reduction trend line for bacterial survival at 12 h compared to 72 h; SA-*Staphylococcus aureus*, EC— *Escherichia coli*, EN— *Enterococcus* sp., SE— *Salmonella enterica*, LM— *Listeria monocytogenes*; (*p* < 0.05).

**Figure 6 ijerph-19-04310-f006:**
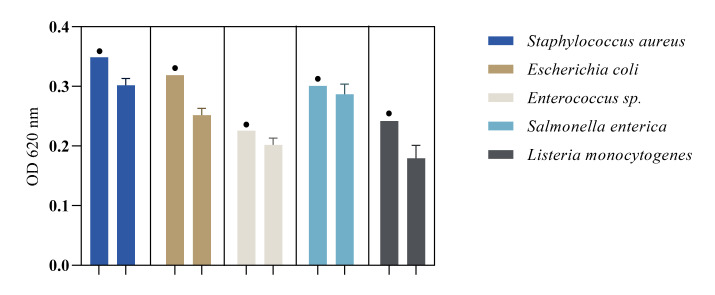
Bacterial recovery rates • = positive control (initial inoculum of 10^5^ CFU/mL in saline buffer) (*p* < 0.05).

**Figure 7 ijerph-19-04310-f007:**
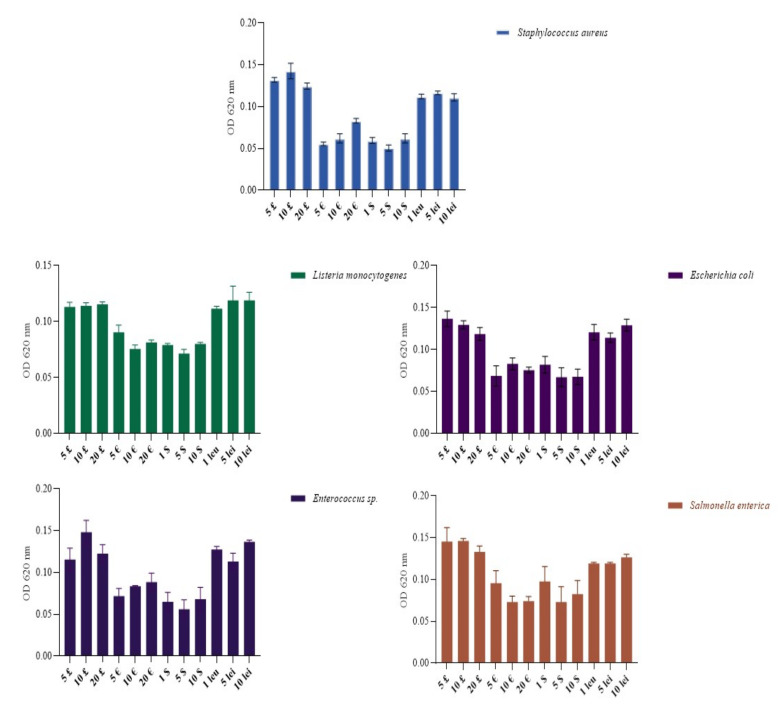
Adherence pattern of *Staphylococcus aureus*, *E. coli*, *Enterococcus* sp., *Salmonella enterica* and *Listeria monocytogenes* strains on different types of banknotes.

**Table 1 ijerph-19-04310-t001:** Preliminary microflora identification.

Sample Type	Colony Aspect on Growth Media	MALDI-TOF Identification
Sample 1 (5 £)	None detected	N/A
Sample 2 (10 £)	None detected	N/A
Sample 3 (20 £)	One colony white creamy aspect	*Micrococcus luteus*
Sample 4 (1 leu)	None detected	N/A
Sample 5 (5 lei)	4 colonies white creamy aspect on Columbia blood agar and CHO medium	*Staphylococcus epidermidis*
	2 colonies white/grey creamy aspect on Columbia blood agar and CHO medium	*Bacillus megaterium* and *Bacillus idriensis*
	1 colony grey growly aspect grew on Columbia agar, CHO and Cled agar	*Fictibacillus arsenicus*
Sample 6 (10 lei)	Numerous grey-glowly, non-hemolytic colonies on Columbia agar, CHO and Cled agar	*Exiguobacterium aurantiacum*
Sample 7 (1 $)	None detected	N/A
Sample 8 (5 $)	1 colony grey glowly aspect grew on Columbia agar, CHO and Cled agar	*Bacillus megaterium*
Sample 9 (10 $)	None detected	N/A
Sample 10 (5 euro)	None detected	
Sample 11 (10 euro)	1 white creamy colony grew on Columbia agar and CHO agar	*Staphylococcus epidermidis*
Sample 12 (20 euro)	None detected	N/A

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
