# Peer review of "Microbial Contamination and Survival Rate on Different Types of Banknotes"

_ijerph, 2022, doi:10.3390/ijerph19074310_

Round 1

Reviewer 1 Report

Current manuscript considers capability of different currency bills (cotton vs polymer-based) to transmit human pathogens. This has been evaluated on several levels. First,  currency was tested for presence of culturable pathogens. Then, bills were sterilized and inoculated with different common pathogens. Survival of pathogens on bills was investigated for 72 hours as well as adherence capabilities of bacteria to the chosen bills. Authors were able to detect some culturable bacteria on investigated bills. A question on whether the presence of colony forming units of some pathogenic bacteria can lead to their transmission to humans has not been addressed in the manuscript. Further, authors have shown that after 72 h of incubation of selected bacteria on the bills, live bacteria are being still detected. This finding does not seem to be surprising, as 72 h is quite a short period of time. But it does prove, that if a bill can be contaminated with bacterial 105 CFU per ml, at least 10 CFUs will survive after 72 h at 22 °C. A question if this might lead to human infections was also not addressed as well as how long one needs to incubate bills, so no viable bacteria are detected any more. Adherence tests have showed that, surprisingly, selected bacteria adhere more to polymer-based bills then to cotton-based. Why that could be the case, authors do not elaborate further. To conclude, current manuscript depicts the problem, that pathogenic bacteria can be detected and can survive on bills of different currency. In context of an ongoing pandemic, when more attention has been given to hygiene, this is an important concern. However, the manuscript does not really depict the research gap and has a number of limitations mentioned below. Besides, manuscript has to be proof-read by a native English speaker. Taking into consideration said above, the reviewer recommends major revision after all the comments and questions will be addressed by the authors.  

Some general comments:

Abstract: a general conclusion is missing.

Introduction: a lot of text is devoted to description of different material used for production of banknotes. How all this information is related to the transmission of bacteria? Please, remove all irrelevant information and before going into details about banknotes, just make a simple introduction, e.g. , in the line 59, that  banknotes can be made from different material, what kind of material and how relevant it is for bacterial transmission. Some examples of already conducted studies on transmission of pathogens and a current research gap would be nice to have.

Material and methods: a lot of details are missing. Methods description sometimes is very sloppy. More consistent explanation of details is needed. E.g., it is not clear how the bacterial survival tests were done. Were contaminated bills put into Mueller-Hinton agar with the contaminated surface downwards?

Results and discussion:

  • correct description of results is missing: for example, if one result is compared to the other, mostly there is no description to which extend the results differ and if this difference is statistically significant. Obviously, a reader is expected to figure it out by himself. But this makes results section very difficult to comprehend. One example: “Currency notes of lower denominations were the most contaminated, presumably because lower denomination bills pass through more hands in 274 their lifetime than the higher denomination bills.” There is no reference to a figure here. There is no mention of a number to what extend bills of lower denomination were most contaminated, what is meant under most contaminated and how this result can be confirmed.
  • Graphical representation of results should be improved: e.g., very similar shades of grey make it cumbersome to identify which bacteria belong to which shade. It would be better to use different colours or different filling options for the bars. Cotton-based currency (euro and dollars) could be plotted next to each other in contrast to leu and pounds (also plotted next to each other) (e.g., Fig. 4).
  • Sometimes there is reference to “previous reports”, but citation of the previous works is missing (e.g. Lines 287-288).

Line 43: is there a new sentence starting with : “Our data showed…?

Line 49: please, add one or two sentences with general conclusion to the abstract.

Lines 130-132: could you, please, explain here more on the connection between spread of viruses and bacteria? Otherwise, a reader is forced to go to the reference material to get more insights.

Lines 135-141: please, use consistent referencing and, please, make sure that sentences are presented in a logic manner. For example, how can “a plastic polymer surface has lower adherence and survival“? Did you mean “bacteria on a plastic polymer surface have lower adherence to this surface and survival rate”?

Line 148: please, give here details, what are those 12 banknotes? How were they obtained? How were they stored? Were replicates presented? Also, please, indicate a reasoning for investigating different denomination bills from one type of currency.

Line 152: what is CHO, Cled ((Cystine Lactose Electrolyte Deficient-Agar)?) agar? Could you, please, give a full name of the abbreviation? Could you, also, please give here the purpose, for the selection of which bacteria these different agar media were used for?

Lines 153-154: could you, please, give more details on how samples were prepared for MALDI-TOF and how analysed, which databases were used?

Line 165: what does it mean just 15-16 hours? Developed on agar? How agar density was adjusted to standard density of McFarland ? Please, give more details.

Line: 168:  CFU instead of UCF

Lines 179: CFU is not a number of bacteria, but a number of colony forming units.

Line 180: Was there negative control?

Line 197: was there negative control? Banknotes could also leach some colour due to the presence of inks, which would interfere with the results!

Line 223: could you, please, state how specific the method used for identification of bacteria on banknotes is? What is about unculturable bacteria? How can they be detected? What could tell the detected bacteria in Table 1? What could be their origin and can their presence on banknotes lead to infection? What could be a probability of such an event?

Line 236: what could be examples of possible resources?

Figures 1 and 2: Misspelling in a title. Please, indicate what bars stand for. Annotation of a y-axis: in the formula for reduction it is said that CFU/ml is calculated. Here y-axis has annotation of CFU/cm2. Please, explain what has been used for the y-axis? CFU or survival rates? Why simply not to use 10, 20 instead of 101 and 2x101?

Line 253: But not only Salmonella or Listeria survived at 72 h, but also all other tested bacteria. Could you, please, explain logic here. Why only these two bacteria are being mentioned?

Lines 273-275: please, explain to which figure you relate to.

Line 277: could you, please, give more details here? Where is indication of 37C in Fig 4? How is it obvious from the figure, that survival rate is dependent on substrate composition? Could you, please, indicate in figures for a readers convenience type of material of banknotes? If you mention, that something is lower or higher in the text, please, indicate to what extend and if it is statistically significant difference.

Figure 5. It is not correct to connect all values with one line, because these values are artificially put together and are not taken from the same experiment.

Line 310, line 317: could you, please, explain why the optical density does not correlate? What were the findings in those reports? Reader is forced again to go to the literature by himself. could you, please, explain what could be a reason for contradiction with already published results? Could absence of a negative control be of a clue here?

Line 314: please, indicate which are Gram positive and which are Gram negative bacteria (for reader`s convenience?

Figures 6 and 7: what is the difference or connection between the figures? Are results in Fig. 7 mainly the same but more detailed results of Fig. 6?

Reviewer 2 Report

Overall, very well-written article. However, the introduction needs to be revised and there is a major concern on the methodologies used in the experimental procedures, which need to be modified and redone. Please read below for more details.    

The last sentence of the introduction states “This contribution is to add to the knowledge of these banknotes serving as fomites in the transmission of bacteria”. However, the major focus of the background is the type and composition of the banknotes (which should be shortened), and very little is said about the significance of increased transmission by fomite as stated in the first sentences of the abstract and the pathogenicity of the bacteria chosen to be tested in this study. The abstract also stated, “Microorganisms were selected in accordance with the criteria of prevalence, pathogenicity, opportunism and incidence” and “common pathogens that are traditionally thought to be found on banknotes” but little to nothing in the introduction referred to this statement.

Line 122 to 129: missing a reference

Line 132: the statement” These bacteria from banknotes are able to be transferred to human skin” should be revised since the previous sentence refers to both bacteria and viruses and the reference to bacteria was not on banknotes. I suggest either removing “these” or adding this information to the previous sentence as “and transferred from banknotes to human skin.”

Line 167: “adjusted to a standard density of 0.5 McFarland. Initial suspensions were then serially diluted to 105 UFC/mL” it is not clear as to what was used to dilute and make the suspension of bacteria. Was saline buffer or broth used for the suspension and if so, then was there any possible growth versus survival only?

There is a major problem with methodology: a sample should have been taken at time zero to be able to compare the number of bacteria at time zero using the same swab collection methodology to the number at time 12, 24,48, and 72 hours. Also, there is no negative control which would be also swabbing the same surface area of the same banknotes that would have been inoculated with only the broth or saline buffer(used for the bacteria but I am not clear as to what was used for the bacteria to make the suspension) without the bacteria.

Line 168: “105 UFC/mL.” should be CFU.

Line 150: states” dry swabs were chosen instead of moist in order to better simulate real handling conditions”. However, this method is assessing for microflora identification, not transmission rate so why not use a moist cotton swab to make sure to collect all bacteria in the square area tested to check for an actual number of bacteria present instead of transferred using a dry object. Also, in the conclusion, it is stated “The low numbers and types of bacteria sampled from collected banknotes were probably due to the use of dry swabs in this study. We would encourage the use of moist swabs or direct contact for future studies. The higher numbers on the local currency may simply indicate greater use and more recent handling”. If the methodology is not reliable, then results should not be reported.

Line 181-197: this method is described as “Evaluation of bacterial adherence”. However, the methodology only describes the evaluation of all bacteria on the banknotes in saline solution compared to “positive control the initial inoculum (105 CFU/mL)” which is not the evaluation of adherence but just persistence or growth. To evaluate adherence, the banknotes should have been washed in PBS first to remove any non-adherent bacteria and then immersed in a saline buffer and shaken vigorously using a vortex. Then the absorbance of the broth, the PBS of the washed, and the saline buffer from vortex to assess the percent adherence versus non-adherence of all bacteria at that time point instead of comparing it to the initial load. This might explain the final findings stated in lines 309-310 “The optical density results obtained for assessment of bacterial adherence capacity 309 on banknotes does not correlate with other reports found [20, 21, 31] relating to banknote 310 material content.” Furthermore, in the conclusion, it is stated” The adherence to the currency did not appear to be strong in the test conditions. 348 Most of the bacteria were recovered from the notes following vigorous vortexing” which is an expected result because the methodology is missing the removal of non-adherent bacteria (washing loose bacteria) before assessing the adherent bacteria. These experiments need to be modified and restarted.

Line 223-225: this sentence is in reference to the preliminary microflora identification and it wasn’t clear in the methods section 2.1 title that it was for the sampling of Preliminary microflora identification.

Reviewer 3 Report

A very topical topic during the pandemic. It is a pity that the authors only examined the survival of bacteria, which do not differ significantly from each other. They are all mesophyles that come in a vegetative form. Instead of Salmonella (its survival in the natural environment is comparable to that of Escherichia coli), it was possible to test bacteria whose survival is much longer, such as Clostridia. It was also possible to use fungal spores, e.g. Candida albicans. In the case of this yeast, the substrate of the notes could be a good medium. Please consider in future research.
 In the conclussion chapter, please add a few sentences of specific conclusions from the research carried out. As such, this chapter is too general.

Round 2

Reviewer 1 Report

Dear authors, thank you for replying to my comments. Now manuscript has become much more clear and structured, I do not have anything to add, except of , there is still a need in some spelling check.

Author Response

Dear Reviewer,

Thank you for your valuable remarks and for your decision.

A full English spell check will be performed by the journal.

Best regards,

The authors